Illusory resizing of the painful knee is analgesic in symptomatic knee osteoarthritis

Stanton Tasha R. tasha.stanton@unisa.edu.au 1 2
Gilpin Helen R. 1
Edwards Louisa 1
Moseley G. Lorimer 1 2
Newport Roger 3 4
1 School of Health Sciences, University of South Australia , Adelaide , South Australia , Australia
2 Neuroscience Research Australia , Randwick , New South Wales , Australia
3 School of Psychology, University of Nottingham , Nottingham , United Kingdom
4 School of Sport, Exercise and Health Sciences, Loughborough University , Loughborough , United Kingdom
Fitzgibbon Bernadette
Electronic publication date: 2018 Jul 17
Publication date: 2018
Volume: 6
Electronic Location ID: e5206
Received 2018 Mar 2; Accepted 2018 Jun 20
Copyright: ©2018 Stanton et al.
Copyright year: 2018
Copyright holder: Stanton et al.
License: This is an open access article distributed under the terms of the Creative Commons Attribution License, which permits unrestricted use, distribution, reproduction and adaptation in any medium and for any purpose provided that it is properly attributed. For attribution, the original author(s), title, publication source (PeerJ) and either DOI or URL of the article must be cited.
License URL: https://creativecommons.org/licenses/by/4.0/

Keywords: Knee osteoarthritis, Bodily illusions, Visuotactile illusions, Multisensory integration, Pain, Mediated reality

Funding: Canadian Institute for Health Research Postdoctoral Training Fellowship ID223354 National Health & Medical Research Council Early Career Fellowship ID1054041 National Health and Medical Research Council Research Fellowship ID1061279 Tasha R. Stanton was supported by a Canadian Institute for Health Research Postdoctoral Training Fellowship (ID223354) and a National Health & Medical Research Council Early Career Fellowship (ID1054041). G. Lorimer Moseley was supported by a National Health and Medical Research Council Research Fellowship (ID1061279). This work was supported by a University of South Australia Internal Research Grant. The funders had no role in study design, data collection and analysis, decision to publish, or preparation of the manuscript.

==============================
Background

Experimental and clinical evidence support a link between body representations and pain. This proof-of-concept study in people with painful knee osteoarthritis (OA) aimed to determine if: (i) visuotactile illusions that manipulate perceived knee size are analgesic; (ii) cumulative analgesic effects occur with sustained or repeated illusions.

Methods

Participants with knee OA underwent eight conditions (order randomised): stretch and shrink visuotactile (congruent) illusions and corresponding visual, tactile and incongruent control conditions. Knee pain intensity (0–100 numerical rating scale; 0 = no pain at all and 100 = worst pain imaginable) was assessed pre- and post-condition. Condition (visuotactile illusion vs control) × Time (pre-/post-condition) repeated measure ANOVAs evaluated the effect on pain. In each participant, the most beneficial illusion was sustained for 3 min and was repeated 10 times (each during two sessions); paired t-tests compared pain at time 0 and 180s (sustained) and between illusion 1 and illusion 10 (repeated).

Results

Visuotactile illusions decreased pain by an average of 7.8 points (95% CI [2.0–13.5]) which corresponds to a 25% reduction in pain, but the tactile only and visual only control conditions did not (Condition × Time interaction: p = 0.028). Visuotactile illusions did not differ from incongruent control conditions where the same visual manipulation occurred, but did differ when only the same tactile input was applied. Sustained illusions prolonged analgesia, but did not increase it. Repeated illusions increased the analgesic effect with an average pain decrease of 20 points (95% CI [6.9–33.1])–corresponding to a 40% pain reduction.

Discussion

Visuotactile illusions are analgesic in people with knee OA. Our results suggest that visual input plays a critical role in pain relief, but that analgesia requires multisensory input. That visual and tactile input is needed for analgesia, supports multisensory modulation processes as a possible explanatory mechanism. Further research exploring the neural underpinnings of these visuotactile illusions is needed. For potential clinical applications, future research using a greater dosage in larger samples is warranted.

Introduction

Knee osteoarthritis (OA) affects 3% of the global population (Cross et al., 2014) and is a condition for which treatment is not always straightforward. Given the discordance between the extent of structural damage on imaging and the extent of joint pain (Hannan, Felson & Pincus, 2000) as well as the occurrence of severe joint pain after total knee replacement (Wylde et al., 2011), it is acknowledged that other neural factors likely contribute to the pain experienced by those with knee OA.

Recent experimental and clinical research has highlighted intriguing links between body representations and pain (Longo et al., 2009; Longo et al., 2012; Mancini et al., 2011; Moseley, Parsons & Spence, 2008; Preston & Newport, 2011). Experimental evidence supports the presence of visually induced analgesia, that is, merely having vision of your own body (versus of an object) reduces pain (Longo et al., 2009). Perceived characteristics of the body also modulate this analgesia, but this effect is not straightforward (Boesch et al., 2016). For example, magnifying the visual input of the hand—making the entire hand appear larger—increases the extent of analgesia in experimental pain (Mancini et al., 2011), but has the opposite effect in pathological hand pain (Moseley, Parsons & Spence, 2008) and has no effect in painful hand OA (Preston & Newport, 2011). Rather, in painful hand OA, combining both touch input and visual manipulation (i.e., a visuotactile illusion: visually increasing the size of the hand while also gently pulling on the fingers) to provide a site-specific change in morphology is analgesic (Preston & Newport, 2011). Specifically, these visuotactile illusions in OA provide a non-affine change in hand morphology. That is, the overall hand does not change size, rather it only ‘stretches’ from one point. Such an illusion which localises its effect to a specific area seems intuitively relevant for a condition such as OA where pain is usually limited to a specific joint. Intriguingly, in hand OA sometimes the analgesic visuotactile manipulation is a stretched looking hand and other times it is a shrunken looking hand (Preston & Newport, 2011), suggesting that the effect might be individually specific.

While some controversy exists regarding the analgesic effects of body illusions (Gilpin et al., 2014; Martini, Perez-Marcos & Sanchez-Vives, 2014b; Mohan et al., 2012), recent work has highlighted that differing effects likely relate to differences in methodology (Martini, Perez-Marcos & Sanchez-Vives, 2014a; Nierula et al., 2017). Indeed, there is a growing body of literature on the theoretical and clinical implications of bodily illusions (Moseley, Gallace & Spence, 2012) and our recent systematic review and meta-analysis highlighted their clear therapeutic potential (Boesch et al., 2016). Importantly, that review also identified the variability of results across methods, experimental, and clinical conditions and emphasised the need for more rigorous and controlled experiments in different types of painful conditions (Boesch et al., 2016). The review also found that most studies evaluated only very small dosages (i.e., the effect of only a single illusion intervention) (Boesch et al., 2016). Evaluating potential cumulative benefit of repeated or sustained illusions is key for clinical relevance.

This exploratory proof-of-concept study aimed to determine whether visuotactile illusions are analgesic for people with painful knee OA. We hypothesised that visuotactile illusions would result in a significantly larger pain reduction than observed during control conditions, although we were uncertain whether vision-only bodily illusions may also provide benefit given previous contradictory findings described above. Last, we explored whether sustained or repeated trials might offer cumulative benefit.

Materials & Methods

Participants

Participants with current knee pain and a clinical diagnosis of knee OA (Altman et al., 1986) were recruited from the community via newspaper advertisements, recruitment posters, and word of mouth. Specifically, if radiographic evidence of osteoarthritic changes were present, then participants were required to have current knee pain and meet one of the following criteria: age >50 years; morning stiffness <30 min; crepitus of the knee. If radiographs were not available, then participants were required to have current knee pain and meet at least three of the following criteria: age >50 years; morning stiffness <30 min; crepitus of the knee; bony tenderness of the tibiofemoral joint line; bony enlargement of the knee; no palpable warmth. Those with rheumatoid or inflammatory arthritis, with neurological disorders affecting the lower limb, or with cognitive impairment were excluded. All participants provided written, informed consent as per the Declaration of Helsinki. This research was approved by The University of South Australia’s Human Research Ethics Board (Protocol No.: 0000028496).

Past work in symptomatic hand OA found large analgesic effects of visuotactile illusions when compared with a control condition (Cohen’s f = 0.6) (Preston & Newport, 2011). Using f = 0.6, power = 0.80, alpha = 0.05, and repeated measures correlation of 0.6, we would need six participants to detect similar effects. Using G*Power 3 (Faul et al., 2007), we conservatively powered to detect a moderate-large (Cohen, 1969) effect on pain (Cohen’s f = 0.35; equivalent to a partial η2 of 0.11), resulting in a required sample of 12 participants.

Equipment

The MIRAGE-mediated reality system (Preston & Newport, 2011) was used to provide two types of visuotactile illusions. One induced a feeling of stretching the knee (stretch illusion) and one induced a feeling of shrinking or compressing the knee (shrink illusion). Illusions were induced with the participant either in sitting or standing. Participants wore a head mounted display that showed a live video feed of their own knee. If tested in sitting, this set-up allowed them to view their knee and leg from a first-person perspective and in the same spatial location as if they were looking down at their own knee (i.e., camera above and slightly behind their head, pointing downwards at their knee). The non-test limb was draped with a black cloth so that participants could only see their test limb. If tested in standing, participants saw their leg in third-person perspective (i.e., camera in front of the leg), but were advised to imagine that they were looking at their own limb that was reflected in a large mirror placed in front of them. Participants stood surrounded by black sheets hanging from ceiling to floor (on the left side, right side and behind them), with the non-test limb covered using a black cloth such that they only had vision of their test limb. In both set-ups participants were familiarised with the technology and underwent a standardised procedure to promote ownership of the limb, i.e., participants moved their legs, flexed their quadriceps muscle and were touched on the leg (∼4 minutes in total), all while watching their own leg in the head mounted display video feed. The choice of illusion set-up (sitting or standing) was determined by which of the two postures was associated with the participant’s typical knee pain. A customised Labview program (National Instruments 2015; Austin, TX, USA) was used to digitally alter the video feed in real-time, such that participants watched their own limb undergo a real-time change in size.

The visuotactile illusions used in this study provide temporally and directionally congruent visual and tactile information to create a sense that the body is truly changing in size (See Video S1 and S2). In the stretch illusion, as the video image of the knee was elongated (making the knee joint appear to stretch or grow), the experimenter applied gentle tactile traction to the participant’s calf muscle (pulling towards the foot) to provide ‘directionally congruent’ information. Similarly, in the shrink illusion, gentle tactile compression (push towards the knee) was accompanied by visual shrinkage of the knee. These manipulations have been found to alter perceptions of body size (Gilpin et al., 2015), reduce pain in people with hand OA (Preston & Newport, 2011) and induce the feeling that the knee is actually stretching or shrinking.

Procedure

Participants attended three sessions. In Session 1, we collected demographic (age, sex, height, weight) and OA-specific information (history of knee pain [years]; minimum, maximum, and average knee pain over the past 48 h using a 0–100 numerical rating scale [NRS], where 0 = no pain at all and 100 = worst pain imaginable). Participants then completed the Oxford Knee Score questionnaire (Dawon et al., 1998) to evaluate knee function, and the Fremantle Knee Awareness Questionnaire (Nishigami et al., 2017) to evaluate body perception related to the knee. Participants completed a perceived knee size experimental task, using established methodology (Gilpin et al., 2015). In brief, participants were presented with a visual image of their own knee that was too small (80%) and too large (120%). These images were increased and decreased in size, respectively, with participants advised to verbally indicate when the image looked to be the right size of their own knee. The order (small/large) was randomised and the procedure was completed twice for each image size presentation (Gilpin et al., 2015).

Following completion of baseline assessment, participants underwent eight conditions in a randomised order (See Fig. 1A): Congruent visuotactile stretch and shrink (as described above); Vision only stretch and shrink (visual image elongates/shrinks; experimenter’s hand on leg but no tactile force provided); Tactile only stretch and shrink (tactile traction/ compression, no visual change); Incongruent visuotactile stretch and shrink (visual stretch, but tactile compression; visual shrink, but tactile traction). For ease of reading, these conditions will be referred to as Congruent VT, VO, TO, and Incongruent VT, respectively. The participants were blinded to condition: no information about the real illusion was provided. Pain intensity, assessed using a 101-point numerical rating scale (where 0 = no pain at all and 100 = worst pain imaginable) was evaluated before and after each condition. Each condition took ∼30 s to complete and there was a 2 minute break between each condition.

Figure 1 Experimental conditions and their statistical comparisons.

(A) The eight experimental and control conditions. The red arrow indicates the direction of tactile input provided. In the Congruent Visuotactile illusion, the tactile input directionally ‘matched’ the visual manipulation (i.e., knee visually shrunk to look smaller, tactile push towards the knee to ‘match’ visual input); in the Incongruent Visuotactile illusion, the tactile input did not directionally ‘match’ the visual manipulation (e.g., knee visually shrunk to look smaller, tactile pull away from the knee, ‘unmatched’ to visual input). Photograph credit: Anne Graham. (B) Statistical comparisons. The grey shaded areas represent the control conditions for which the most analgesic congruent visuotactile illusion was compared to for analysis purposes.

Following application of the eight conditions, the congruent VT illusion that resulted in the greatest pain reduction immediately post-illusion was then applied for a second time and sustained for 3 minutes while participants viewed their knee in this altered state. Pain intensity was reported every 30 s during this 3 minute period. The total duration of Session 1 (including baseline questionnaires; See Table 1) was approximately 1 hour.

Table 1 Participant demographic and testing session outcomes.

All pain outcomes measured using a 101-point NRS. Oxford knee scores range from 0–48 where higher values indicate less disability. Knee awareness/perception was evaluated using a modified version of the Fremantle Knee Awareness Questionnaire (FreKAQ); scores range from 0–36 with higher scores reflecting less knee awareness (Nishigami et al., 2017). Perceived knee size was evaluated using established methodology (Gilpin et al., 2015) : a picture of a participant’s knee was altered in size; participants indicated when the viewed image appeared to be the correct size of their knee.

	Mean (SD)	
Demographics		
Age (years)	67.3 (9.9)	
Gender (count)	9 female	
Height (cm)	167.2 (11.2)	
Weight (kg)	82.7 (16.3)	
Bilateral painful knee OA (count)	6	
History of knee pain tested knee (years)	16.5 (14.3)	
History of knee pain untested knee (years)	7.0 (5.4)	
Average baseline knee pain (past 48 hrs)	48.0 (24.3)	
Maximum knee pain (past 48 hrs)	66.3 (28.6)	
Minimum knee pain (past 48 hrs)	6.3 (10.9)	
Oxford knee score	24.1 (8.1)	
Knee awareness/perception (FreKAQ)	14.0 (8.4)	
Perceived knee size (% of true size)	104.0 (0.05)	
Session 1		
Visuotactile illusion resulting in the most analgesia (count)	stretch –7; equivocal –2; shrink –3	
‘Best’ illusion (visuotactile or visual only)	visuotactile –9; visual –3	
Sustained illusion:		
Post-illusion pain (directly after)	28.5 (17.0)	
Sustained: post-illusion pain (180 s)	26.4 (18.9)	
Session 2		
Repeated illusions:		
Pre-illusion 1 pain	31.7 (12.9)	
Pre-illusion 10 pain	21.7 (17.5)	
Post-illusion 1 pain	23.3 (8.8)	
Post illusion 10 pain	17.2 (16.6)	
Session 3		
Sustained illusion:		
Post-illusion pain (directly after)	27.4 (15.5)	
Sustained: post-illusion pain (180 s)	28.4 (17.7)	
Repeated illusions:		
Pre-illusion 1 pain	50.4 (24.6)	
Pre-illusion 10 pain	31.3 (22.0)	
Post-illusion 1 pain	42.3 (23.1)	
Post illusion 10 pain	30.4 (21.3)	
Daily pain scores		
48 h after Session 2	45.1 (16.8)	
48 h before Session 3	58.1 (25.2)	

Sessions 2 and 3 (minimum of two weeks apart; maximum of three weeks) used the illusion that was determined most analgesic during Session 1. In Session 2, the effect of a 3 minute sustained illusion was evaluated again (assessing pain every 30 s). In Sessions 2 and 3, 10 trials of the illusion were performed, assessing pain intensity pre- and post-illusion. During testing with the congruent VT illusion, participants were asked whether or not it felt as though the manipulation was occurring to their own leg (yes/no). This question was not asked during the eight conditions of Session 1 in order to maximise participant blinding to the ‘real’ illusion. Last, participants recorded their average daily pain scores between the second and third session using a pain diary.

Statistical analysis

All statistics were performed using IBM SPSS 22.0. Data were assessed for normality (using visual inspection and Shapiro–Wilk statistic) and for sphericity (using Mauchly’s test of sphericity). If the normality assumption was not met for raw nor for transformed data, non-parametric analyses were used. If the sphericity assumption was violated, Greenhouse-Geissier corrections were applied.

Our pilot data in those with knee OA (n = 3) showed that one type of congruent VT illusion (e.g., stretch) was more analgesic than the other illusion (e.g., shrink) and control conditions; but whether the analgesic illusion was stretch or shrink varied between participants. Thus our analysis plan, determined a priori, identified the congruent VT illusion (stretch or shrink) that was most analgesic in each participant and compared pain ratings with those of the relevant control conditions (See Fig. 1B).

To determine if the congruent VT illusion provided analgesia above that provided by its component parts (VO, TO) we performed a 2 (Time: pre-/post-condition) × 3 (Condition) repeated measures analysis of variance (RM ANOVA). Post-hoc paired t-tests were used to explore any significant effects, using a Holm-Bonferroni correction (Holm, 1979) to control for multiple comparisons. To determine if the congruent nature of visuotactile input was important, we performed a 2 (Time) × 2 (Condition: congruent vs incongruent) RM ANOVA. Separate analyses were completed to compare to each incongruent condition (i.e., vision-controlled and touch-controlled). Additionally, given that frame of reference (first-person versus third-person perspective) has been shown to play a critical role in the phenomenal experience of body ownership and the effectiveness of experiencing bodily illusions (Blanke, 2012), we repeated the above analyses, including Perspective (first- vs third-person; relating to whether the participant was seated or standing during testing, respectively) as a between subject factor. Because we did not initially power for this analysis, it was considered exploratory.

Given that visual distortion of body size alone (i.e., no tactile component) is analgesic (Mancini et al., 2011), and that varying effects are seen between individuals with chronic pain (Preston & Newport, 2011) we performed a supplementary analysis comparing pain scores (pre-/post-condition) from the ‘best’ condition with those for the relevant tactile control condition. The ‘best’ condition was considered the most analgesic of either VO or congruent VT conditions.

Last, to determine if sustained illusion or multiple trials offered extra benefit, paired t-tests compared pain intensity: (i) immediately post-illusion versus end of 3 minutes (sustained); (ii) the 1st illusion versus the 10th illusion (repeated). To determine if there was a decrease in average daily pain (last 48 h), paired t-tests compared the baseline measures of average pain with the average pain directly after the 2nd session and prior to the 3rd session (the latter two involving the average of the daily pain scores for two days).

Results

Fourteen participants were screened for inclusion; two were ineligible because they did not meet ACR criteria for knee OA (Altman et al., 1986). Both ineligible participants did not have radiographs of their knee and neither satisfied at least three of the six clinical criteria. One participant had knee pain following total knee replacement surgery but was included because it mirrored the original osteoarthritic knee pain. Twelve participants completed Session 1; six completed Session 2 (three had no pain in sitting/standing, and thus could not be tested; three dropped out due to time commitments and reported difficulties getting to the testing lab); seven completed Session 3 (two participants had no pain). During sustained and repeated testing with the congruent VT illusion, all participants reported that it felt as though the manipulation was occurring to their own limb (for both testing set-ups: 6 –sitting; 6 –standing). Full participant demographics are provided in Table 1. All outcomes were normally distributed and thus parametric statistics were used.

Effect of congruent VT illusion versus control conditions Fig. 2A

The congruent VT illusion resulted in significantly more analgesia than the TO and the VO control conditions (Fig. 2A). There was no effect of Condition (F2,22 = 0.93, p = 0.41), no effect of Time (F1,11 = 4.7, p = 0.053), but a Condition x Time interaction (F2,22 = 4.2, p = 0.028). Paired t-tests showed no change in pain during the TO (t1,11 = 1.45, p = 0.17) and VO control conditions (t1,11 =  − 0.71, p = 0.95), but a significant pain reduction during the congruent VT illusion (t1,11 = 2.96, p = 0.013). Pain decreased by an average of 7.8 points (95% CI [2.0–13.5]), corresponding to a 25% reduction from pre-illusion pain scores. Considering perspective type (first- vs third-person) in the analysis did not change the findings and there was no main effect of Perspective or any of its interactions (see File  S1).

Figure 2 Pre-/post-condition pain scores comparing experimental conditions.

Pain intensity was rated on a 0–100 NRS where 0 = no pain at all and 100 = worst pain imaginable. *p < 0.05; N.S. = non-significant (A). Mean pre- and post-condition pain scores (±SEM) for comparisons between the Congruent VT illusion and its components: vision only control, tactile only control. A significant Condition x Time interaction was found; post-hoc comparisons showed that the congruent VT illusion provided significant analgesia, while both component conditions did not. (B) Mean pre- and post-condition pain scores (±SEM) for comparisons between the Congruent VT illusion and the Incongruent VT Conditions. Separate repeated measures ANOVAs showed a main effect of Time (pre-/post-) when the visual manipulation was identical (i.e., tactile input differed) in Congruent and Incongruent conditions, but no effect when the tactile input was identical (i.e., visual manipulation differed) in Congruent and Incongruent conditions.

The congruent VT illusion pain ratings did not differ from the incongruent VT condition that controlled for vision (Fig. 2B). That is, when identical visual manipulation occurred, there was a main effect of Time (F1,11 = 12.6, p = 0.005), but no effect of Condition (F1,11 = 0.032, p = 0.86), or Condition × Time interaction (F1,11 = 0.34, p = 0.57), suggesting that analgesia was provided by both conditions. These findings were unchanged when considering first- versus third-person perspective and there was no main effect of Perspective or any of its interactions (See File S1). In contrast, the congruent VT illusion resulted in significantly more analgesia than the incongruent VT condition that controlled for tactile input. That is, when identical tactile input occurred, there was no effect of Condition (F1,11 = 0.73, p = 0.41), and a main effect of Time (F1,11 = 5.23, p = 0.043), driven by a Condition × Time interaction (F1,11 = 5.29, p = 0.042) whereby the incongruent VT condition (touch controlled) did not result in analgesia (t1,11 = 1.26, p = 0.23). Again, these findings were largely unchanged when considering first- vs third-person perspective, the only difference found was that the Condition × Time interaction became non-significant (p = 0.052), although this is likely due to reduced power (See File  S1).

The exploratory analysis comparing the ‘best’ condition (congruent VT or VO) to the TO control condition found similar findings (no effect of Condition, F1,11 = 1.10, p = 0.32; main effect of Time, F1,11 = 14.4, p = 0.002; Condition × Time interaction, F1,11 = 10.6, p = 0.008), but enhanced analgesia. Paired t-tests showed a significant reduction in pain for the ‘best’ illusion (t1,11 = 4.2, p = 0.002), with an average pain reduction of 11.9 points (95% CI [5.6–18.2]), corresponding to a 37% reduction in pain. There was no change in pain for the TO condition (t1,11 = 1.5, p = 0.17).

Effect of sustained illusions on pain (Table 1)

There was no additional analgesic effect of sustained viewing of the congruent VT illusion, but the initial effect was sustained. Pain intensity immediately after the illusion did not differ from pain intensity after 3 min of sustained viewing of the illusion (Session 1: t1,10 = 0.52, p = 0.61; Session 3: t1,7 =  − 0.697, p = 0.51).

Effect of repeated illusions on pain

In Session 2, pain scores for congruent VT illusion one did not differ from illusion ten (pre-illusion scores: t1,5 = 1.4, p = 0.21; post-illusion scores: t1,5 = 1.1, p = 0.33). However, in Session 3 pain scores following illusion ten were significantly reduced compared with pain scores for illusion one (pre-illusion: t1,6 = 3.5, p = 0.013; post-illusion: t1,6 = 3.9, p = 0.008; Fig. 3). The analgesic effect was large: a reduction of 20 points (95% CI [6.9–33.1]) from the 1st to 10th illusion, corresponding to a 40% reduction in pain.

Figure 3 Pre- and post-illusion pain scores over 10 repeated illusions.

Planned comparisons performed between illusion 1 and 10, show that 10 repeated illusions significantly reduce both pre-illusion and post-illusion pain. ∗p < 0.05

Effect of illusions on daily pain scores (Table 1)

There was no difference between average knee pain (last 48 h) at baseline and average daily pain in the 48 h after Session 2 (t1,6 = 0.54, p = 0.61) or the 48 h prior to Session 3 (t1,6 =  − 1.31, p = 0.24). However, average daily pain scores (last 48 h) were significantly lower directly after Session 2 than those taken just prior to Session 3 (t1,6 = 2.70, p = 0.036).

Discussion

We found evidence that illusory knee resizing using visuotactile manipulation is analgesic in people with osteoarthritic knee pain. Congruent VT illusions reduced pain, while the individual touch and vision components did not, suggesting that pairing of sensory input is important to the analgesic effect. Contrary to our hypothesis, whether or not the tactile input ‘directionally matched’ the visual input appeared less important. Congruent VT illusions were not more effective at reducing pain than incongruent VT conditions that involved identical visual input (but opposite tactile input), but were more effective than incongruent conditions that involved identical tactile input (and opposite visual input). This suggests that vision is critical to the effect, but requires the pairing of multisensory input (i.e., tactile) to alter pain. Last, prolonged viewing of the illusion sustained analgesia, but did not increase its magnitude. Repeated application of these illusions increased analgesia, but may require a larger dosage than 10 illusions to achieve the added benefit. Daily pain scores were not affected by this brief experimental dosage.

Analgesic differences between congruent VT illusions and the visual and tactile components

Our results in those with knee OA support past work showing that bodily illusions can modulate clinical pain (Boesch et al., 2016). That congruent VT illusions were analgesic and that separate visual and tactile components were not, suggests the presence of a super-additive effect on pain during the VT illusion. Such effects are the hallmark of multisensory integration, classically demonstrated by behavioural and perceptual responses that exponentially improve with multisensory versus unisensory input (Stein & Stanford, 2008). Greater analgesia with congruent VT illusions than tactile input alone (TO condition) suggests that changes in nociceptive drive (via traction/pressure changes) or gating at the spinal cord via tactile input (Kakigi & Watanabe, 1996), are unlikely to contribute to the effect observed. Vision of the body and visual resizing of the body have analgesic effects in experimental pain (Longo et al., 2009; Longo et al., 2012), but, consistent with findings in hand OA (Preston & Newport, 2011), we did not see such an effect here.

Comparison of analgesic effects between types of illusion (affine versus non-affine illusion)

It is also interesting to consider the impact of the type of illusion provided—that is, whether it was affine (i.e., a rigid body transformation that magnified or minimized the entire body part) or non-affine (i.e., a non-rigid body transformation that altered only part of the body part). Past work has shown that visual illusions that magnify the overall size of the hand (i.e., affine illusions) have contradictory effects on pain dependent upon the condition. For example, magnifying the hand reduces experimental pain (Mancini et al., 2011), but increases pain in those with pathological limb pain (Moseley, Parsons & Spence, 2008), and has no effect in people with painful hand OA (Preston & Newport, 2011). However, non-affine alterations that provide site-specific visual morphology changes (e.g., stretch or shrink illusions) are analgesic in hand OA (Preston & Newport, 2011). The present work shows that non-affine alterations to the knee are also analgesic in people with knee OA. Further it extends past work by showing that non-affine visual only change (no tactile component) does not modulate pain. Given past findings of no effect on pain of overall hand size visual change (affine) in people with hand OA (Preston & Newport, 2011), our results support the view that the type of illusion (affine vs non-affine) and the components of the illusion (i.e., visuotactile) are key to the analgesic effect in painful OA.

Differing amounts of analgesia induced by illusion for hand and knee OA

That congruent VT illusions provide analgesic benefit in knee OA is consistent with findings in hand OA (Preston & Newport, 2011); however, the magnitude of effect seen here was not as large (25% vs 45% pain reduction, respectively). There are several potential explanations for this difference. First, various studies show that tactile input from the hand is more precisely represented in the primary somatosensory cortex (S1) than tactile input from the knee (Catley et al., 2013; Mancini et al., 2014; Penfield & Boldrey, 1937). Given that body resizing illusions are thought to target brain-held body maps (see Schaefer et al., 2007 for evidence of S1 changes with altered visual input of arm size), this less precise cortical representation of the knee might at least partly explain the differing responses to VT illusions and therefore the size of the analgesic effect.

Second, it may be that body-specific multimodal integration of vision and touch (a hypothesised mechanism, via S1 inhibition (Cardini, Longo & Haggard, 2011), for pain modulatory effects of visuotactile illusions), that occurs in the superior colliculus (Stein, Standford & Rowland, 2014), the premotor area and the posterior parietal cortex (PPC) (Avillac et al., 2004; Bremmer et al., 2001), may differ based on bodily site. Studies of multisensory illusions show fundamental differences in the process of multisensory integration in the lower versus upper limbs, with the legs appearing less sensitive to sensory inputs (Pozeg, Galli & Blanke, 2015; Van Elk, Forget & Blanke, 2013). Further, that we spend a great deal of time throughout our development watching our hands closely as we manipulate objects, would suggest that visuotactile representations of the hands may be more efficacious and sensitive than those of the knee. Together these findings would support a reduced analgesic effect in the knee versus the hand.

Third, illusory resizing inherently results in a spatial incongruence between the visually perceived size, and the actual size, of the body part. The impact of this incongruence on pain may differ between the hand and the knee. In the hand, incongruence between body-specific information and spatial information (i.e., crossing the hands over midline) is analgesic, and this effect occurs in later stages of processing of the nociceptive signal, which is thought to coincide with integration of body relevant information in the PPC (Gallace et al., 2011). Such incongruence may impair multisensory processing, thus modulating pain. However, in the lower limb, multisensory integration is not modulated by limb crossing (Van Elk, Forget & Blanke, 2013), therefore it is possible that analgesic effects induced by impairments in multisensory processing are not present.

Spatial incongruence of viewed and actual body size as an analgesic mechanism for body illusions is not straightforward. Many people with chronic pain have been shown to have distorted perceptions of the size of their painful body part (Lewis & Schweinhardt, 2012; Moseley, 2005) (see Moseley, Gallace & Spence, 2012 for review)—including those with OA (Gilpin et al., 2015; Nishigami et al., 2017). Incongruence between predicted and actual movement (heightened by inaccurate perceptions of the body) may be algesic in some conditions (McCabe, Cohen & Blake, 2007; McCabe et al., 2003), although see also Moseley & Gandevia (2005). It is interesting to consider whether illusions may normalise pain-induced distortions in bodily size perception—that is, does changing the perceived size of the knee actually reduce body-specific incongruence because the brain-held perception of its size is already inaccurate? On average, participants overestimated the actual size of their knee (Table 1; perceived knee size). This might suggest that they would prefer a shrink illusion: if mental representation is too large then showing a smaller knee to shrink the representation should be best (i.e., visually normalising size). However, one may argue that if a body part is perceived as being too big, then an illusion that matches the visual size that the body is expected to be may be analgesic (i.e., visually matching expected size). Indeed, most participants responded to illusory stretch (only three to illusory shrink), which may support the latter idea. Regardless, that most participants responded to illusory stretch (despite overestimating their knee size) may not be inconsistent with a hypothesis of normalising body perception. Past work has shown that regardless of the type of illusory manipulation (stretch or shrink), distortions in perceived hand size in people with hand OA normalise to that of hand size chosen by healthy pain-free volunteers (Gilpin et al., 2015). Clearly further work is needed to disentangle such effects.

Importance of visuotactile input, but not directional congruence

That incongruent and congruent VT illusions provided equivocal analgesic effects, but only when the same visual manipulation occurs, suggests that visual input is critical. But visual input alone (i.e., VO) is not sufficient to produce analgesia, highlighting that multisensory input (i.e., tactile) is required to influence pain. Why might this be? It is possible that inclusion of tactile input (regardless of directional congruence with vision) increases the sense of ownership—the feeling that this is happening to ‘my leg’. Experimental pain models support that increases in ownership (of a rubber hand) are analgesic (Siedlecka et al., 2014). However, this hypothesis remains speculative given that we did not formally evaluate ownership for each of the experimental conditions in Session 1.

Effects on multisensory integration in the lower limb may also occur without the need for congruent tactile input. For example, having a first-person viewpoint during lower limb illusions (i.e., congruent vision and proprioceptive input) increases visuotactile integration, but congruent tactile input (i.e., synchronous vs asynchronous tapping) does not provide an additional effect (Van Elk, Forget & Blanke, 2013). While some participants had a third-person viewpoint during illusions, our supplementary analysis evaluating the effect of viewpoint on pain ratings showed no effect of first- versus third-person perspective. While this analysis may be underpowered to detect such an effect, it may also be that our strategy of having participants imagine that they had a large mirror in front of them (and thus were viewing a mirrored image of their limb), resulted in assuming a first-person perspective. Past work has shown that perceiving a mannequin as mirrored when viewing it in a third-person perspective results in similar levels of ownership as when viewing the mannequin in a first-person perspective (Preston, Kuper-Smith & Ehrsson, 2015).

Last, perhaps vision overrides tactile input. Given that vision provides us with (usually) reliable and precise sensory information about our body, it may be that the increased precision of visual input is sufficient to dominate direction information from tactile input (i.e., we do not detect directional incongruence). Our work shows that vision is heavily weighted when judging the location of our body (i.e., the hand), even when proprioception provides contradicting, and accurate, information (Bellan et al., 2015). Past work has shown that incongruent VT input (using the rubber hand/full body illusion set-up) does not induce an illusory percept and is not associated with strong multimodal integration due to spatiotemporal incongruence between visual and tactile input (Blanke, Slater & Serino, 2015). Thus it may be argued that multimodal integration and the generation of an illusory percept does not underlie the analgesic effects seen here given that the incongruent VT condition also provided analgesia. However, it is important to note that asynchronous simulation can induce ownership during the rubber hand illusion in some people (albeit not as strongly as synchronous stimulation) (Botan et al., 2018; Rohde, Di Luca & Ernst, 2011) and that the sense of ownership in multisensory illusions is additionally sensitive to cues about visual appearance and spatial location when those illusions are applied to one’s own body (Ratcliffe & Newport, 2017). Further, the present incongruent VT condition may be argued to have spatial congruence (the researcher’s hand is located and tactile input occurs at the visually seen location on the leg) and temporal congruence (tactile input occurs at the same time the visual change occurs), but not tactile directional congruence (the tactile input does not match the direction of visual change). Given some level of spatiotemporal congruence, this then may allow visual input to take priority and multimodal integration to occur despite some level of directional incongruence. Future work is clearly needed to delineate the mechanism by which this analgesic effect occurs.

Sustained versus repeated illusions

That sustained illusions did not increase analgesic benefit, but that repeated illusions did, suggests that the analgesic effect may be driven by neural processing initiated with viewing the real-time change in body size. Motion is known to capture visual attention (Abrams & Christ, 2003); it is possible that this could be one explanation as to why repeated moving illusions appear to have larger analgesic effects than sustained, static illusions. However, static images of magnified hands are analgesic in experimental pain (Mancini et al., 2011) suggesting that analgesic effects are not solely due to motion in the illusion (and may explain why sustained viewing of the illusion sustained analgesia). It is also possible that relative imprecision in visual representations of the knee may result in participants rapidly adopting the sustained illusion as being an accurate reflection of their own knee, thus not triggering additional analgesic effect as the condition is maintained. Unsolicited comments from participants support this—many remarked that during the sustained illusion, they no longer felt that their knee was resized. On the contrary, when illusions are repeated, cueing of a change in knee size is repeatedly provided, thus potentially re-instating modulatory processes driven by vision.

Study limitations

Our study recruited a small sample but conservative, a priori power calculations based on past work (Preston & Newport, 2011), suggest that it was adequately powered. Sessions 2 and 3 had lower participant numbers, however, this is unlikely to affect the results—sustained illusions were completed in the full sample in Session 1 with identical findings to that of Session 2 and the effect of repeated illusions on pain was large, and significant, in Session 3. While the effects of VT illusions on pain were small (∼8 points on a 101-point NRS), these relate to a single 5-second illusion; repeated illusions resulted in pain relief of 20 points, which notably meets recommendations for a clinically important difference (Farrar et al., 2001; Salaffi et al., 2004).

In our analysis of daily pain scores, we compared a retrospective recall at baseline (i.e., average pain over the last 48 h) to the mean of two days of average daily pain scores taken post-Session 2 and pre-Session 3. Past work has shown good agreement between the recall of average pain over the last two to seven days and daily pain scores in OA (Nguyen, Yeasted & Schnitzer, 2014; Perro et al., 2011), with low levels of error (Giske, Sandvik & Roe, 2010), providing confidence in our findings. However, this is a potential limitation. We do note that daily pain ratings were assessed identically between Session 2 and 3. Further research on the effect of these illusions on daily pain scores is warranted.

A final limitation is that we did not take ratings of the vividness of the congruent VT illusion (and control conditions) during application or of the general ownership that participants had of the viewed limb during each condition. This was a pragmatic decision made to maximise participant blinding for condition (e.g., participants may be more aware of the ‘real’ illusion if we cued them to these features). We informally evaluated ownership of the VT illusion that was tested during sustained and repeated illusions (Sessions 1 to 3) by asking participants if what they experienced felt like it was happening to their own limb and all participants reported that it felt like it was their own limb that was changing (irrespective of perspective). This area is clearly ripe for future research, for example, to elucidate whether vividness of the illusion relates to the degree of analgesia and whether or not ownership also occurs during incongruent VT illusions.

Conclusions

This study adds to the existing evidence suggesting that manipulation of body-relevant sensory information has a modulatory effect on pain. Our results extend previous work by showing that pain modulation by illusory resizing also occurs in knee OA and by clearly demonstrating that the visual component of the congruent VT illusion is critical but requires multisensory input to have an analgesic effect. Such results warrant replication in a larger sample, providing a greater dosage to ascertain whether daily pain scores can be impacted.

Supplemental Information

Supplemental Information 1 Raw data for Sessions 1–3

Click here for additional data file.

File S1 Exploratory analyses: the effect of illusion perspective (first vs third) on pain outcomes

Click here for additional data file.

Video S1 Visuotactile congruent knee illusions in first person perspective—stretch

The video illustrates the first person perspective that participants received when tested in a sitting position. This perspective was used when the normal knee osteoarthritic pain was present in a sitting (non-weight bearing) position.

Click here for additional data file.

Video S2 Congruent visuotactile knee illusion in third person perspective

The video illustrates the third person perspective that participants received when tested in a standing position. Participants were asked to imagine as though they were looking at their own limb in a large mirror placed in front of them. This perspective was used when the normal knee osteoarthritic pain was present only in a weight bearing (standing) position.

Click here for additional data file.

We would like to thank Dr Valeria Bellan for her initial help in the piloting phase of this study.

Additional Information and Declarations

Competing Interests

Author Contributions

Human Ethics

Data Availability

Tasha R. Stanton received travel and accommodation support from Eli Lilly Ltd. for speaking engagements (September 2014) on these topics. G. Lorimer Moseley receives royalties for books on pain and rehabilitation and speaker’s fees for lectures on these topics. He has received support from Pfizer, workers’ compensation boards in Australia, Europe and North America, Kaiser Permanente (USA), Arsenal Football Club, the Port Adelaide Football Club, and the International Olympic Committee. Roger Newport is the creator of the MIRAGE mediated-reality systems; The University of Nottingham received equipment fees for creation of systems for external labs.

Tasha R. Stanton conceived and designed the experiments, performed the experiments, analyzed the data, contributed reagents/materials/analysis tools, prepared figures and/or tables, authored or reviewed drafts of the paper, approved the final draft.

Helen R. Gilpin conceived and designed the experiments, performed the experiments, authored or reviewed drafts of the paper, approved the final draft.

Louisa Edwards analyzed the data, prepared figures and/or tables, authored or reviewed drafts of the paper, approved the final draft.

G. Lorimer Moseley conceived and designed the experiments, authored or reviewed drafts of the paper, approved the final draft.

Roger Newport conceived and designed the experiments, contributed reagents/materials/analysis tools, authored or reviewed drafts of the paper, approved the final draft.

The following information was supplied relating to ethical approvals (i.e., approving body and any reference numbers):

The University of South Australia’s Human Research Ethics board granted ethical approval to carry out the study within its facilities (Ethical Application Ref: 0000028496).

The following information was supplied regarding data availability:

All raw data have been uploaded as a Supplemental File.

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
