# Peer review of "Illusory resizing of the painful knee is analgesic in symptomatic knee osteoarthritis"

_PeerJ, doi:10.7717/peerj.5206_

## Round 0.1 · original submission · Minor Revisions

· Academic Editor

Minor Revisions

As noted, the reviewers are enthusiastic about the manuscript but have made several suggestions or queries. Your paper should become acceptable for publication pending suitable revision and modification of the article in light of the appended reviewer comments.

·

Basic reporting

The manuscript entitled with “Illusory re-sizing of the painful knee is analgesic in symptomatic knee osteoarthritis” by Stanton et al. is clearly structured and written in professional and unambiguous language. To my opinion, the main findings are sufficiently emphasized. The proof-of-concept study provides a novel treatment option based on a body part illusion employing multimodal integration with bodily signals. The augmented-reality application sounds promising and after a well-applicable and affordable method in everyday clinical practice. The authors found a significant pain reduction only using the multimodal visuotactile illusion. The authors found increased analgesia with repeated trials of the visuotactile illusion, however, the authors critically discuss that a potentially to low dosage was the reason for not observing sustained pain reduction in daily pain ratings. The cumulative analgesic effects are important for clinical practice.
However, there are certain aspects mainly concerning shortages in the methods description and the experimental design that shall be addressed by the authors.

Experimental design

A. First-person vs third-person perspective:
I suggest reporting on the additional factor of looking at the knee from a first-person perspective vs. a third-person perspective. The frame of reference (e.g., egocentric vs. allocentric) is a crucial factor determining the effectiveness of bodily illusions and the phenomenal experience of ownership (Blanke, 2012). The authors report that some subjects performed the conditions from a first-person perspective while the others performed the task from a third-person perspective in the method section. Potential differences between both viewpoints were not addresses later on. I suggest reporting on potential differences, since the authors also speculate that the sense of ownership might be an important mediator of the effectiveness of the VT illusion in the discussion part without showing data on ownership experiences during the conditions.

B. Baseline pain description in the methods section:
The authors report on daily pain assessments between session 2 and session 3 and that statistical comparison were performed between the baseline pain and the pain directly after session to and prior to session 3. However, from the methods section it is not obvious to me when the baseline pain measurement was assessed and whether it was also assessed outside of the laboratory environment and thus similar to the pain diary assessments. The first description of the baseline is given in the results section in Line 206 as far as I can see.

C. No ratings on the strength and vividness of the VT illusion and pain descriptives
The authors discuss (speculate) that sense of ownership might be an important aspect of the treatment efficacy. However, the authors do not show results demonstrating the intensity of the illusion and whether it was linked to the analgesic effect.

D. Rigid-body vs. non-rigid body transformation of visualized body parts:
The motivation of using stretching and shrinking visuotactile illusions rather than magnifying and minimizing illusions could be emphasized and contrasted more. The authors compare on pain modulation via shrinking/stretching, while former literature (also referenced by the authors) refer to magnifying and minimizing illusions. While the former describes affine (rigid-body) transformations in shown body parts the latter refers to non-affine alterations in visualized body sizes. The article on shrinking vs stretching illusions (Gilpin et al., 2015) only looked at the presence of malformed hand representation in hand OA but not on illusion-related alterations in pain.

E. Further points:
I was unable to find the total duration of session 1 as well as the duration of the eight individual conditions.

It was not obvious to me if the subjects were able to see the other leg. Was the other leg visually occluded? I suggest addressing these points in the methods section.

The authors commonly refer to illusion conditions, however the VO, TO and incongruent VT are expected to not elicit illusory precepts. I suggest substituting ‘illusion’ by ‘condition’ if necessary.

I suggest rephrasing pre-/post-illusion pain scores to pre-/post-condition pain scores since the control conditions do not elicit an illusory percept.

Validity of the findings

The authors do not report on whether the assumptions to perform repeated measures ANOVAs were fulfilled (e.g., test of sphericity, normality…).

Due to high drop-out rates in session 2 and session 3 linear mixed models might be a fruitful alternative to repeated measures ANOVA due to their ability to deal with outliers fairly well.

The authors do not clearly state that they performed a priori contrasts in the methods section (e.g., Line 145) but report on planned contrasts in the Figure description 2. Thus, I strongly recommend explicitly mentioning in which cases planned contrast or otherwise multiple comparisons control were performed in the methods section.

The authors report that sometimes the shrinking and sometimes the stretching congruent VT illusion was most analgesic. It would be interesting to know whether the verbal description of pain is qualitatively related to the higher effectiveness of either the stretch or shrink illusion. For instance the McGill Pain Questionnaire assesses whether pain is perceived to be ‘pressing’ or ‘pulling’. Do the authors have data on how the patients describe their OA knee pain?

The authors conclude that multimodal integration and the generation of an illusory re-sizing of the knee is a key aspect of the immediate and prolonged pain reduction. But, the authors also report that the congruent and incongruent VT conditions were most effective at reducing pain. The incongruent VT illusion is, however, expected to not induce an illusory percept and not to be associated with strong multimodal integration due to spatiotemporal incongruence between visual and tactile input (Blanke et al., 2015). Moreover, some of the participants performed the VT conditions from a third person perspective, which might have lowered the sense of ownership additionally. Thus I am not fully convinced that multimodal integration and illusory re-sizing precepts are the key aspect of the therapeutic pain reduction found. Do the authors have data on the illusory strength in the incongruent VT condition to address that point?

Additional comments

Figures and Tables
Figure 2:
• I recommend writing ‘A significant Condition x Time interaction was found’
• I suggest reporting on the verbal endpoints of the 0-100 NRS
• I recommend an indication of what the error bars represent (M ± CI, SE or SD).
Figure 3:
• I recommend rephrasing the title to […] over ten repeated illusion trials.
• I recommend rephrasing to […] between the first and tenth illusion trial, […]
• I suggest reporting on the verbal endpoints of the 0-100 NRS
• I recommend an indication of what the error bars represent (M ± CI, SE or SD).
Table 1:
• I recommend indicating verbal endpoints of scales if existing
Abstract
Line 28-29: I suggest speaking about body representations rather than brain-based body representations since (1) no brain imaging was performed and (2) mental as well as neural body representations support a link to pain.
Line 34: I suggest highlighting the verbal endpoints of the 0-100 numerical ratings scale.
Introduction
Line 61-62: Please add references.
Line 63: I suggest introducing the terminology of ‘visually induced analgesia’ (Longo et al., 2009)
Lines 78-79: I recommend rephrasing to ‘the effect of only a single illusion intervention’ since one illusion might be confused with one type of illusion.
Line 88: I would prefer to read a short description of the diagnosis criteria on knee OA.
Materials and Methods:
Line 88: I recommend referring to participants or subjects rather than people.
Line 121: visual image elongate(s)/shrinks
Line 128:
a. Since the control conditions did not induce an illusion, I suggest rephrasing to ‘[…] was evaluated before and after each condition.’
b. Rephrase: between each test condition  confusing since most conditions served as control conditions.
Line 133-134: ‘was evaluated again’ rather ‘was again evaluated’.
Line 147: the abbreviation ‘RM ANOVA’ was not introduced beforehand. In line 145 repeated measures ANOVA is used.
Line 157: change 1st to 1st.
Results
Line 163: I suggest reporting on the ACR criteria shortly. Abbreviation ACR not introduced here or in the methods section.
Line 241: the abbreviation S1 was not introduced.
Line 250: the abbreviation PPC was not introduced.
Line 316: Please add a reference: … past work (…).
Line 320: the abbreviation NRS was not introduced.
Line 403: Please change ‘..Pain’ to ‘.Pain’
Line 418: Please change ‘.. Eur J Pain’ to ‘. Eur J Pain’
Inconsistencies:
• resize vs. re-size
• VT illusion vs. VT illusions
• p = XXX vs. p=XXX
• I suggest spelling in full numbers between zero and twelve.

LITERATURE
Blanke O. Multisensory brain mechanisms of bodily self-consciousness. Nat. Rev. Neurosci. 2012; 13: 556–571.
Blanke O, Slater M, Serino A. Behavioral, Neural, and Computational Principles of Bodily Self-Consciousness. Neuron 2015; 88: 145–166.
Gilpin HR, Moseley GL, Stanton TR, Newport R. Evidence for distorted mental representation of the hand in osteoarthritis. Rheumatology (Oxford). 2015; 54: 678–82.
Longo MR, Betti V, Aglioti SM, Haggard P. Visually induced analgesia: seeing the body reduces pain. J. Neurosci. 2009; 29: 12125–12130.

·

Basic reporting

The paper is written very well, with sufficient references. Note that there has been some controversy regarding body illusions and analgesia - see
http://www.jpain.org/article/S1526-5900(17)30017-2/abstract
https://onlinelibrary.wiley.com/doi/full/10.1002/ejp.606

In the abstract the authors refer to "brain-based body representations". I don't understand the 'brain-based' part of this. Aren't all body representations 'brain-based'? I think that the 'brain-based' can be left out. Perhaps 'illusory body representations' might be preferable, since that is what is being dealt with.

The paper is self-contained, and follows the expected structure.

Experimental design

The research is certainly within the aims and scope of the journal, being an original study with a well-defined research question that provides new information about the field of research.

The work is carried out to a high standard.

The setup with respect to participants being seated or standing is not clear. I don't understand why if they would stand it would be equivalent to looking in a mirror. This wasn't clear from the video, which, I think only had the sitting down condition.

Validity of the findings

The analysis has been carried out satisfactorily. On line 94 the authors refer to an 'f' value. Presumably this is an effect size, but it needs to be specified which one. I think most researchers will know Cohen's d, but there are many others, and it isn't obvious what 'f' refers to.

Following the repeated measures ANOVA the authors use paired t-tests (e.g., line 145, 155, 173, ...). I think that these have not been controlled for multiple comparisons. There are very many tests in this paper, so some attention to the issue of multiple comparisons would be useful. I would avoid the Bonferroni which is extremely conservative. The authors might like to consider this approach, which is simple to do and has had more than 40,000 citations:
http://www.stat.purdue.edu/~doerge/BIOINFORM.D/FALL06/Benjamini%20and%20Y%20FDR.pdf
However, I'm not insisting on this, but the number of comparisons could be a problem.

Additional comments

This paper adds usefully to the literature, and apart from the minor points above is ready for publication.

---

## Round 0.2 · accepted · Accept

· Academic Editor

Accept

The authors have addressed the reviewers concerns.

# ·

Basic reporting

The concerns mentioned in my minor revision were fully met by the authors.
The shortages in the description of the illusion induction as well as the statistical analysis were met by the authors.
The manuscript reads very well.

Experimental design

The authors give a sufficient description of the experimental paradigm. Moreover, the shortages in the description of the statistical analysis (esp. regarding multiple comparisons control) were compensated.


Typo in Line 143: Greenhouse-Geissier --> Greenhouse-Geisser

Validity of the findings

The authors included a statistical comparison to detect potential differences between inducing the illusion from a third-persons versus first-person perspective, which helps understanding the mechanistic basis of the illusion.

·

Basic reporting

All comments have been appropriately addressed.

I found this typo: lines 32-33
"recent work has highlighted that that differing effects likely relate to differences in methodology"

Experimental design

Previously discussed in first review.

Validity of the findings

Previously discussed in first review.

Additional comments

Previously discussed in first review.